# Chemical and Biological Components of Urban Aerosols in Africa: Current Status and Knowledge Gaps

**DOI:** 10.3390/ijerph16060941

**Published:** 2019-03-15

**Authors:** Egide Kalisa, Stephen Archer, Edward Nagato, Elias Bizuru, Kevin Lee, Ning Tang, Stephen Pointing, Kazuichi Hayakawa, Donnabella Lacap-Bugler

**Affiliations:** 1Institute for Applied Ecology New Zealand, School of Science, Auckland University of Technology, Auckland 1142, New Zealand; ekalisa@aut.ac.nz (E.K.); kevin.lee@aut.ac.nz (K.L.); donnabella.lacapbugler@aut.ac.nz (D.-L.B.); 2School of Sciences, College of Science and Technology, University of Rwanda, P.O. Box 4285, Kigali, Rwanda; ebizuru@gmail.com; 3Institute of Natural and Environmental Technology, Kanazawa University, Kakuma-machi, Kanazawa, Ishikawa 920-1192, Japan; nagatogou@se.kanazawa-u.ac.jp (E.N.); n_tang@staff.kanazawa-u.ac.jp (N.T.); hayakawa@p.kanazawa-u.ac.jp (K.H.); 4Yale NUS-College and Department of Biological Sciences, National University of Singapore, Singapore 138527, Singapore; stephen.pointing@yale-nus.edu.sg

**Keywords:** polycyclic aromatic hydrocarbons, nitrated polycyclic aromatic hydrocarbons, microorganisms, particulate matter, carcinogenic

## Abstract

Aerosolized particulate matter (PM) is a complex mixture that has been recognized as the greatest cause of premature human mortality in low- and middle-income countries. Its toxicity arises largely from its chemical and biological components. These include polycyclic aromatic hydrocarbons (PAHs) and their nitro-derivatives (NPAHs) as well as microorganisms. In Africa, fossil fuel combustion and biomass burning in urban settings are the major sources of human exposure to PM, yet data on the role of aerosols in disease association in Africa remains scarce. This review is the first to examine studies conducted in Africa on both PAHs/NPAHs and airborne microorganisms associated with PM. These studies demonstrate that PM exposure in Africa exceeds World Health Organization (WHO) safety limits and carcinogenic PAHs/NPAHs and pathogenic microorganisms are the major components of PM aerosols. The health impacts of PAHs/NPAHs and airborne microbial loadings in PM are reviewed. This will be important for future epidemiological evaluations and may contribute to the development of effective management strategies to improve ambient air quality in the African continent.

## 1. Introduction

Africa has the fastest growing population in the world and this is predicted to more than double between 2017 and 2050 [1]. This rapid population growth is associated with greater industrialization, motorization, and urbanization, creating dense urban centers. As a result, emissions from internal combustion engines, domestic cooking, and open fires contribute to worsening air quality. Air pollution is the single largest cause of premature human mortality worldwide. Annually, ~4 million people die prematurely from illness attributable to household biomass smoke such as pneumonia, chronic obstructive pulmonary disease (COPD), and lung cancer [2]. The impacts disproportionately affect the world’s poorest, most vulnerable populations. This is particular concern in sub-Saharan Africa, where people are still heavily reliant on biomass fuel for cooking or heating, with the lowest proportion (5%) of people using clean fuel as primary source of energy [3]. In 2016, the World Health Organization (WHO) reported that household air pollution in Africa contributed to almost 739,000 deaths [4]. According to the Organization for Economic Co-operation and Development (OECD), air pollution in Africa will be the highest cause of environmentally related deaths by 2050, overtaking unsafe water and poor sanitation [5]. While these figures are alarming, there is little data on air quality and health-related problems. The extent to which an individual is harmed by air pollution depends on their total exposure to pollutants, including a measure of the duration of exposure, the concentration of pollutants, and population vulnerability. Thus, the evaluation of particulate air pollution for a country is typically measured by PM_2.5_ (particulate matter with aerodynamic diameter less than 2.5 micrometers) that can enter the lungs and PM_10_ (particulate matter with aerodynamic diameter less than 10 micrometers) that are trapped in the nasopharyngeal tract [6]. In 2013, ambient air pollution of PM was classified by the International Agency for Research Cancer (IARC) as a group I carcinogen [7]. Exposure to high levels of PM_2.5_ and PM_10_ have been identified as causes of cancer, asthma, pulmonary fibrosis, oxidative stress, immune response, and chronic obstructive pulmonary disease [8].

The development of coal-fired industries and increased automobile use have overlapped, which has resulted in the emissions of a complex mix of air contaminants [9]. Recent evidence suggests that PM is a mixture of chemical and biological origin [10,11]. The total PM includes biological organisms (e.g., bacteria, fungi, and viruses), organic compounds (e.g., polycyclic aromatic hydrocarbons (PAHs) and their nitro-derivatives (NPAHs)), nitrates, sulfates, metals (e.g., iron, copper, nickel, zinc, and vanadium), and elemental carbon [12]. These components vary substantially according to time, location, season and climate, which results in spatial–temporal variation in characteristics, concentration, and toxicity [13,14,15,16]. PM-bound PAHs and NPAHs are the most studied components as they were found to be carcinogenic and enhance mutagenic properties [17,18,19,20]. A review on PAHs and their association to cancer revealed that there was an increase in lung cancer (relative risk of 1.2–1.4) and bladder cancer (relative risk of 2.2) in occupationally exposed subjects (40 years of exposure) [21]. Previous studies demonstrated that almost a quarter of the total airborne PM above land surfaces is made up of biological material [22,23]. Another study indicated that the chemical composition of PM could provide insight into a variety of problems related to PM emissions [24]. As particles of both biological and chemical origins are transported together with air currents in the atmosphere, PM can be used as a carrier of both pathogenic microorganisms—bacteria, fungi, and viruses [25]—and carcinogenic compounds of organic aerosols [12,26]. Depending on their concentration and meteorological factors [22], inhalation of these mixtures can have significant effects on the health of the population (Figure 1).

The health effects of airborne PM have been linked to its chemical and biological components [27], while its interaction with regard to composition is influenced by meteorological conditions (long range transport, temperature, and relative humidity) and the physical properties of the PM (Figure 1) [28]. Recent evidence suggests that there may be an association between chemical and biological components in PM size fractions that result in increased negative health outcomes [12]. Boreson et al. [29] and Skóra et al. [30] indicated that toxic chemical particles could be used as a carrier of other pathogenic microorganisms and such interaction would have serious implication, as biological components could conceivably be penetrating deeper into the lungs than would have been expected. Consequently, health effects, such as COPD, asthma, and lung cancer (Figure 2), may be enhanced when biological and chemical components in PM are combined together [29]. However, the association of these factors is complex and requires comprehensive research.

In Africa, the available data on ambient PM levels are generally above the WHO’s annual and 24-h mean guideline value for PM_2.5_ and PM_10_ [31,32]. A study indicated that PM encompasses many different chemical and biological components, which have been cited as major contributors to its toxicity [33]. However, there are still limited studies in Africa on the characterization of chemical and biological components of PM. The few studies that have assessed the chemical components of ambient PM in Africa have demonstrated that PM concentrations and lifetime cancer risks resulting from inhalation exposure to PM chemical composition exceeded WHO safe limits and provide clear evidence that an immediate development of emission control measures is required [34]. The available data from biological composition associated with atmospheric PM comes mostly from Asia [35,36,37], Europe [11], and the United States [38]. However, data on microorganisms associated with PM are scarce for Africa. Understanding both components of PM is crucial, as the relative harm of each component may differ by concentration or composition and the combination of both chemical and biological components may be more harmful than their individual components [12,39]. Evidence suggests that the composition of emissions is more important than merely controlling the absolute sources [33,40]. As such, an inclusive control of all sources of the most toxic air pollutants is called for, together with a robust regulatory framework based on scientific evidence. This review summarizes the association between biological and chemical components of PM and their associated health outcomes, with an emphasis on the Environmental Protection Agency’s (EPA) 16 priority-listed polycyclic PAHs [41], their NPAHs, and pathogenic microorganism loadings in the PM fractions.

A literature search was conducted in academic online databases, such as Web of Science, Google Scholar, American Chemical Society (ACS), PubMed, ProQuest, and Science Direct. Due to the paucity of air quality data in Africa, there was no restriction in literature search in terms of study period and publication date. We searched literature using the MESH terms “PM”, “particulate matter”, “air pollution”, “urban air quality”, “fine particles”, “coarse particles”, “PM_10_”, “PM_2.5_”, “Bioaerosols loading in PM”, “Bacteria associated PM”, “Fungi loading in PM”, chemical composition of PM, “Polycyclic aromatic hydrocarbon in PM”, “PAH”, “nitro-polycyclic aromatic hydrocarbon in PM”, “PAH”, “NPAH”, “carcinogenic PAH” “Organic carbon loading in PM”, and “Human Health effects of PM”. These MESH terms were sometime combined with name of African region or African countries. For chemical composition of PM, we selected studies conducted in ambient air. Data from recognized organization such as WHO, World Bank, OECD, United States Environmental Protection Agency (US EPA), and United National Environmental Program (UNEP) were also included. In the assessment of the health outcome of exposure to air pollution on population study, we selected only studies that use statistic to test exposure response relationship between measured ambient PM and any health outcome of interest. We have also considered studies that assess potential risk of exposure to atmospheric PM-bound-PAHs and NPAHs.

## 2. Overview of Ambient Particulate Matter in Africa

In most African countries, PM pollution is above the annual and 24-h mean air quality guideline value recommended by the WHO [31,32,42] and ambient PM was classified among the top 10 risk factors in sub-Sahara African countries [43]. Despite this, little data and no standards exist for the majority of African countries. Recent studies have shown that one in eight premature deaths globally can be linked to poor air quality and approximately 90% of these deaths occur in low and middle-income countries [44].

In this review, we have selected only publications carried in Africa with actual ambient PM measurements and that show mean PM_10_ and PM_2.5_ levels. Studies that are most recent and that have reported actual mean of PM, sampling device, sampling duration, and sampling site characteristic (traffic roadside, urban background, and rural sites) are shown in Figure 2. A summary of epidemiological studies conducted in Africa on health effects of exposure to a mass concentration of ambient particulate matter size fraction are shown in Table 1. In Appendix A, we provide a broad range of all the references selected with detailed information of methodology used, which, we think, may be of interest to aerosols studies and may be important resources as guiding exemplars for the various strategies, which can be employed when assessing health outcome of PM.

Most of the studies were conducted in urban areas, with a high population density and a large number of potential pollution sources. Additionally, poor logistics on long-term measurement were prevalent as studies were undertaken for less than six months or for less than a 24 h day period; focusing on near-roadway air pollution, using limited equipment, and low flow rate, making comparison of the PM data across studies conducted in Africa difficult [31]. Limited epidemiological studies have been conducted in Africa, and available information on air quality and health was reported by WHO based on satellite data estimates. Four reviews on air pollution studies in Africa have been published [31,32,42,67]; however, only one among these four studies addresses the health effects of air pollution in sub-Saharan countries [42]. Three reviews highlighted that biomass burning and road traffic are major source of high levels of PM pollution in Africa [31,32,67]. Additionally, the study carried in two low-income neighborhoods in Accra, Ghana indicated that combustion source (biomass and traffic) and noncombustion sources (geological and marine) were major contributors of PM pollution [49]. Further, Coke and Kizito [42] have recently reviewed ambient air pollution and health effects in Africa and indicated that population from sub-Saharan Africa are exposed to both acute and long-term health effects from ambient air pollution and highlighted gap in epidemiological studies due to lack of long term PM monitoring.

## 3. Chemical and Biological Components of Particulate Matter Worldwide

### 3.1. Chemical Components of Airborne Particulate Matter 

Substantial improvements have been achieved in chemical characterization and identification of the main PM components in developed and developing countries [17,68,69]. Chemical components of PM typically contribute an average of 20% to the total PM mass load [70]. These components are primarily emitted into the atmosphere while some are formed in the atmosphere. Studies in the United States indicated that airborne PM contains a variety of microorganisms, some of which are pathogenic and pose severe threats to human health [25,71]. However, the chemical composition of atmospheric PM is not distributed equally among all size ranges [72], meaning that chemical composition depends on the aerosol sources. PAHs and NPAHs are known for their harmful health effects, referring to a large group of organic compounds with two or more fused aromatic rings [73]. In the atmosphere, PAHs (two or three rings) exist in the vapor phase, whereas multiringed PAHs (five rings or more) exist in particles phase [74,75]. PAHs are also capable of being transported from one region to another (intercontinental transport long-range transport) via air currents [76]. In addition, more than 90% of the carcinogenic PAHs appear to exist in the particulate phase of ambient air [75]. In this section, the general overview of PM-associated PAHs and NPAHs and airborne microorganism loadings in PM are extensively reviewed. Findings show that airborne microorganism and organic aerosols (PAHs and NPAHs) are associated with PM and may provide reliable data for studying the response of the human body to increasing levels of air pollution.

#### 3.1.1. Particulate Matter-Associated Polycyclic Aromatic Hydrocarbons and their Nitro-Derivatives

PAHs and NPAHs are ubiquitous environmental organic pollutants, which originate from the pyrolysis of organic matter and incomplete combustion of coal, oil, petrol, and wood [74]. NPAHs can form as secondary compounds through atmospheric reactions between PAHs and atmospheric oxidants such as ozone and nitrate radicals [77]. Some PAHs and NPAHs have carcinogenic and/or mutagenic properties, like benzo[a]pyrene (BaP) and 1-nitropyrene (1-NP), which are classified as Group 1 PAH and Group 2A (probably carcinogenic to humans) NPAH, respectively [78]. In addition, several other PAHs and NPAHs are classified in Group 2B (possibly carcinogenic to humans) [79]. Given their toxicity and their wide distribution in the atmosphere, the EPA has classified 16 PAHs as priority compounds [41].

#### 3.1.2. Toxicity of Polycyclic Aromatic Hydrocarbons and their Nitro-Derivatives

Inhalation of PM, including PM_2.5_ and PM_10_, causes respiratory, cardiovascular, and lung diseases such as asthma, COPD, and lung cancer [80]. In China, COPD was reported as the most common cause of human mortality resulting from exposure to high levels of particulate air pollution at home [81]. As PAHs and NPAHs are the major components of PM_2.5_ and PM_10_, they are thought to be responsible for these respiratory diseases [82].

NPAHs, which exist at concentrations orders lower than PAHs, are receiving particular attention since they possess a higher direct-acting mutagenicity and carcinogenicity than PAHs that first undergo an enzymatic activation process [83,84]. In 2013, research published by Pham et al. [85] analyzed the mutagenicity of PMs, PAHs, and NPAHs by the Ames test using *Salmonella typhimurium* strains. PAHs such as BaP and benzo[b]fluoranthene (BbF) were found to cause indirect-acting mutagenicity of PMs exhausted from coal burning, wood burning and automobiles [84,86]. Benzo[a]pyrene is the most widely studied PAH as it can be used as a marker for carcinogenic risk levels in environmental studies [87]. The WHO and several countries including the United States and China have recognized BaP as epidemiological health hazard and have set protective health standards of 1 ng/m^3^, 0.25 ng/m^3^, and 10 ng/m^3^, respectively [88]. NPAHs have also been previously observed in the organic extracts of ambient PM [82]. For example, NPAHs such as 1-nitropyrene (1-NP) and 1,3- and 1,8-dinitropyrenes (1,3-, 1,6-, and 1,8-DNPs) showed very strong direct-acting mutagenicity of emission extracts of diesel engines and wood particulates [86,87,89]. The latter NPAH exhibited high direct-acting mutagenic potency in the Salmonella bacterial mutagenicity assay, and on human lung tissue. Hayakawa [75] indicated that the metabolites of PAHs and NPAHs exhibited estrogenic and antiestrogenic activity in the yeast two-hybrid assay system using yeast cells expressing estrogen receptor.

### 3.2. Biological Components of Airborne Particulate Matter

Biological aerosols are composed of all biologically derived pathogenic or nonpathogenic matter, live or dead, and include bacteria, fungi, and viruses [26]. The size distributions of bioaerosols vary considerably by type: pollens are typically 5–100 μm, fungal spores are 1–30 μm, and bacteria are 0.1–10 μm, while viruses are generally smaller than 0.3 μm [26]. For example, biological aerosols represent a significant fraction of airborne PM and affect the microstructure and water uptake of aerosol particles. Bioaerosols such as bacteria, fungi, and viruses have been shown to account for a significant proportion of the mass of coarse (PM_10_) and fine (PM_2.5_) particles. Airborne bioaerosols may be found as individual particles or agglomerates of particles [90]. It has been that the dynamics of biological particles in the air is governed mainly by the particles’ physical characteristics, of which size and concentration are the most important [27]. Bioaerosol components, such as bacteria, fungi, and viruses, can attach to PM from varied sources from biomass, soil, and industries. Consequently, PM-associated bioaerosols can enhance their penetration into deeper parts of the lungs [91]. For example, a pollen grain (>10 µm) is trapped in the nasopharyngeal tract when inhaled, whereas, pollen allergens present in PM_2.5_ can easily penetrate deep into lungs [26]. As a result, the agglomeration of bioaerosols and PM can exacerbate respiratory allergies and other health ailments such as pulmonary disease, cardiovascular disease, and cancer [12,91].The association and interaction of microorganisms and microorganism-derived allergens with airborne particulates has been documented to be part of the urban aerosphere [37]. Further, bioaerosols has also been identified issues in relation to agricultural and human health [92]. For example, through air dispersal during agricultural activity, many plant pathogens can travel from one region to the other and cause disease outbreaks, leading to severe crop losses, famine, and mass migration [93]. The transport of bioaerosols and other air pollutants in gas phase are influenced by several factors including temperature, relative humidity, wind speed, and physical properties of the bioaerosols [37,91,94]. Davis [95] indicated that *Peronospora tabacina* (blue mold), which is an agriculture disease that caused epidemics in United States tobacco in the late 1900s, is transmissible through the atmosphere. Airborne transmission is also one of the common ways for spreading of infectious human diseases. For example, people working or living in the same environment may spread diseases such as measles, winter stomach flu, influenza, and tuberculosis [96]. Exposure to bioaerosols in the occupational environment is associated with a wide range of health effects with major public health impacts, including infectious diseases, acute toxic effects, allergies, and cancer [97]. 

#### 3.2.1. Particulate Matter-Associated Airborne Fungi 

Fungi originate from natural (plant, animals, and soil) and anthropogenic activities [98]. Fungal spores are a reported threat to human health [99]. Studies indicated that fungal spore are emitted in the atmosphere and become the most dominant biological components in airborne PM [98,99]. Fungi, like pollen and spores, account for large proportions of airborne PM, but some other components such as fungal spores belong to fine fractions of PM [100,101]. A previous study, carried in Australia, showed that PM could attach to fungal spores and airborne pollen and possibly change their morphology [102]. For example, PM of similar size to fungal spores emitted in an atmosphere may coagulate, and their penetration into the human respiratory system may cause more serious implications than they would have otherwise been expected to cause alone [26]. Studies also indicate that fungal spores and pollen contribute 4–11% of the total mass concentration of PM_2.5_ [23]. The concentration of fungal spore loadings on PM is higher in PM_10_ than in PM_2.5_ air samples. This is most likely because the aerodynamic diameters of a fungal spore agglomerate are between 2.5 μm and 10 μm [103]. For example, Cao et al. [104] found fungal spores to be the most common biological components of airborne dominant microorganism loadings in PM_10_, being 4.5% more than in PM_2.5_ (1.7%).

Exposure to fungal spores loading in PM has been associated with respiratory diseases’ allergies and asthma [105]. Several studies have found that *Cladosporium* sp., *Aspergillus* sp., *Penicillium* sp., and *Alternaria* sp. are the most predominant genera of fungi identified in airborne PM samples and they have been associated with symptoms of respiratory tract allergies [29,105,106]. Additionally, tree and grass pollens and fungal spores have been shown to exacerbate respiratory diseases such as asthma and rhinitis [90,105].The protection of sensitive populations from pathogenic fungi requires an understanding of environmental exposures to airborne fungi as a function of type and size.

#### 3.2.2. Particulate Matter-Associated Airborne Bacteria

Airborne bacteria are one of the major components of indoor and outdoor aerosol particles [107,108]. Airborne bacteria can be found in the air as isolated microorganisms but are more likely to be attached to other particles such as soil or leaf fragments, or found as conglomerates of a large number of bacterial cells [26]. Some studies have shown a continuous interaction between the concentration of dust particles and microorganisms [109]. In an urban environment, high concentrations of airborne bacteria can have substantial effects on human health as pathogens or as triggers of asthma and seasonal allergies [107]. For example, several studies have shown that higher biological components in the air, associated with PM, increased both asthma and allergic reactions [108,109]. Several studies have also demonstrated that airborne bacteria are associated with small size particles [13,110]. Nasir and Colbeck [111] proved that up to 80% of the total viable concentration of bacteria (5036 CFU/m^3^) in the atmosphere is found in particles with diameters less than 4.7 μm. Microbial allergens and pathogens were identified in PM, and their relative abundance appeared to increase as the concentration of PM pollution increased [104]. Cao et al. [104] found that the representation of pathogens identified within the entire bacteria community was 0.012% in PM_2.5_ and 0.017% in PM_10_ samples and their concentration appeared to have increased by 2 times from an average of 0.024%.

#### 3.2.3. Particulate Matter-Associated Airborne Viruses

Bioaerosols also consist of viruses responsible for various diseases that affect the public health [26,104]. Due to its small size, the virus can remain airborne, come into contact with humans or animals, and potentially cause an infection. Studies have detected viruses in airborne biological contaminants, despite their small size and the difficulty in collecting and analyzing procedures [102,106]. Cao et al. [104] employed metagenomic methods to analyze the microbial composition of Beijing’s PM pollutants and show that airborne dsDNA viruses can be identified at the species level. They found that Human adenovirus C (6.5%) was the most dominant pathogenic airborne virus identified in the PM_2.5_ and PM_10_ samples. Liang et al. [112] found significant association between ambient PM_2.5_ concentrations and virus (Human influenza) in Beijing, which have important implications for public health and environmental actions. Exposure to airborne viruses plays an important role in microbial ecology and some infectious diseases. Studies have shown an association between viruses and bacteria that cause respiratory infections in children with asthma; additionally, Pneumococcus bacteria and influenza virus have been shown to interact with each other [108,113].

## 4. Chemical Composition of Ambient Particulate Matter in Africa

### 4.1. Atmospheric Concentrations of Polycyclic Aromatic Hydrocarbons and Their Nitro-Derivatives in Africa

In Africa, urbanization and population growth have increased rapidly in recent decades. African countries account for more than a quarter of global energy consumption, with wood burning being the main energy source [114,115]. The burning of these solid fuels and biomass releases several air pollutants, gases and particulates, with two of the toxic organic compounds present in PM_2.5_ and PM_10_, PAHs, and NPAHs being of greatest environmental health concern due to their carcinogenicity and mutagenicity. The majority of studies reviewed in this study found that the mean concentrations of atmospheric PM_2.5_ and PM_10_ in Africa greatly exceeded the 2006 WHO guideline value of annual and 24-h mean, and those carcinogenic and mutagenic organic pollutants are a major component of PM (Figure 2). Despite available publications of atmospheric NPAHs in Rwanda [34], Egypt [116], and Algeria [117] (Figure 3), the atmospheric concentrations of total PAHs and NPAHs show a large variation among these countries. The PAH concentrations, in descending order, were Senegal, Kenya, South Africa, Mali, Uganda, Rwanda, Sierra Leone, Algeria, and Egypt. The NPAH concentrations, in descending order, were Rwanda, Algeria, and Egypt (Figure 3). It must be emphasized that total PAH concentrations in Senegal, Kenya, and South Africa were much higher than those in the remaining countries, suggesting that the urban atmosphere in Senegal, Kenya, and South Africa were much more polluted with PM-containing PAHs [118,119,120]. 

### 4.2. Source and Risk Assessment of Particulate Matter-Bound Polycyclic Aromatic Hydrocarbons and Their Nitro-Derivatives in Africa

NPAHs are formed from PAHs in the presence of nitrogen oxides at high temperature; this means that the corresponding PAHs increases with increasing temperature [86]. The combustion temperatures in wood stove, coal stove, and diesel engine are different [82]. Thus, the concentration ratios of NPAHs to PAHs have been widely used by several authors worldwide, and suggest that [NPAH]/[PAH] ratios are useful markers to identify source type of PAHs and NPAHs For example, The [NPAH]/[PAH] concentration ratios were previously determined in three different types of PM; Diesel engine vehicles (combustion temperature ~2700–3000 °C), coal-burning stoves (~900–1200 °C), and wood burning stoves (~500–600 °C) [86]. For example, the [1-nitroperylene]/[pyrene] or [1-NP]/[Pyr] ratios of coal emissions (0.001) were much smaller than the ratio of diesel emission particles (0.36), and the [1-NP]/[Pyr] ratio was recommended as a marker for source identification of PAHs and NPAHs [86]. African automobile emissions (Diesel and gasoline) and biomass burning were considered major contributors of PAHs and NPAHs in urban and rural sites, respectively (Table 2). The [1-NP]/[Pyr] ratios in Kigali, Rwanda were 0.05 (dry season) and 0.04 (wet season) [34], while the values in the Great Cairo Area, Egypt were 0.06 (winter) and 0.03 (summer) [116]. The values in these two African countries were similar to those reported in East Asian cities influenced by large volumes of vehicle emissions [82]. 

High concentrations of benz(g,h,i)perylene (BPe), phenanthrene (Phe), fluoranthene (Flu), BaP, and benzo(b)fluoranthene (BbF), account for a large proportion of the total PAH, that have been commonly observed in ambient particulates from available studies in Africa (Table 3). High emissions of BPe and indeno (1,2,3-cd)pyrene (IDP) have been associated with vehicle emissions [123], while high emissions of Flu, BaP and BbF are associated with domestic fuel burning [124,125].

NPAHs compounds, such as 9-nitroanthracene (9-NA) and 1-NP directly emitted from Diesel engines, were most abundant NPAHs detected in African cities (Table 2). However, most of the sampling sites in Africa were near the intersection of high traffic volumes, suggesting that these NPAHs were emitted from automobiles. Additionally, several PAH pairs, such as [fluoranthene]/([pyrene] + [fluoranthene]) or [Flu]/([Pyr] + [Flu]), [benz(a)anthracene]/([chrysene] + [benz(a)anthracene]) or [BaA]/([chrysene(Chr)] + [BaA]) and [indeno (1,2,3-cd)pyrene]/([benz(g,h,i)perylene + [indeno (1,2,3-cd)pyrene]) or [IDP]/[BPe + IDP], have been also used as markers of the source of the PAHs in African countries, including Rwanda [34] and Kenya [118]. Biomass burning and automobile emissions were the main sources of atmospheric PAHs in Kenya and Rwanda (Table 2). To evaluate cancer risk of PAHs and NPAHs detected in airborne PM, the methodology developed by the US EPA was widely applied [126,127]. In African countries such as Rwanda (Table 3) findings from cancer risk assessment studies reported that PAHs and NPAHs present in PM_2.5_ and PM_10_ were above WHO recommended health standard (1ng/m^3^) and would be classified as definite risk [126]. Table 2 summarizes the available information on PAHs and NPAHs compounds analyzed in PM size fraction, location, sources, observed health effects, and the details of cited references in Africa.

## 5. Current Understanding of Bioaerosols Associated Particulate Matter in Africa

Africa produces more than 50% of the airborne particles produced worldwide, and millions of metric tons of African desert dust is transported by natural atmospheric processes as far as the United States and Europe [129]. Despite recognition of the importance of bioaerosols from African desert dust, the effects of exposure to these aerosols on humans have never been investigated. African dust contains pathogenic biological particles which have been documented to exacerbate respiratory and proinflammatory diseases [129]. Bioaerosols, their effects on human health, and their long-range transportation from Africa have been extensively studied worldwide [25,52,54,130,131,132,133]. Investigation of the microbial content of African desert dust and related impact on humans is still in its infancy. Even though earlier work in Africa assessed bioaerosols in hospital rooms (Table 3), typical outdoor exposure level to bioaerosols is still unknown. Considering the impact of bioaerosols on human health, examining outdoor bioaerosol exposure levels in different locations and their spatial variability is important to sensitive populations. African dust can significantly increase ambient PM levels contributing excessive amounts of PM as set by the WHO. This is a result of a lack of valid quantitative exposure assessment methods. Characterization of bioaerosol samples is challenging and requires powerful analytical tools and knowledge of molecular biology and aerobiologic chemistry. In Africa, funding for the installation of bioaerosol sampling and air analysis is inadequate. As a result, this limits bioaerosol studies across the region with the few that have been undertaken and completed in collaboration with international institutions.

Early studies in Africa employed cultivation approaches to assess the diversity and composition of airborne bacteria and fungi associated with PM in Egypt, Libya, and South Africa (Table 3). However, these studies provide a limited insight into airborne bacteria and fungi associated with PM, as only viable and culturable microorganisms can be identified through culture methods. Only one study in South Africa that investigated the transmission of *Mycobacterium tuberculosis* applied advancements in enumerating various culture-independent (high-throughput DNA sequencing) techniques [134]. The latter techniques reflect the diversity of airborne fungi and bacteria since they are very sensitive and significantly quicker than traditional methods. This process can be applied to any biological sample containing nucleic acid, as they detect viable, nonviable, culturable, and nonculturable organisms [36,104,135]. A few studies that have been completed in Africa on bioaerosols associated with indoor PM have found that *Bacillus* sp., *Cladosporium* sp., *Aspergillus* sp., and, *Penicillium* sp. are the most predominant genera of bacteria and fungi identified in airborne PM samples. These organisms have been associated with symptoms of respiratory tract allergies, asthma, and infections in patients (Table 3). Bacillus and Staphylococcus have been observed to dominate the bacterial aerosol community in the indoor air samples in Africa [136,137] and some species of bacteria *Acinetobacter calcoaceticus* and *Corynebacterium aquaticum*, known as human pathogens, have been found as airborne in Bamako, Mali [120].

## 6. Conclusions

In Africa, rapid population growth, industrialization, motorization, and urbanization encourage the development of dense urban centers and contribute to the worsening air quality. Further to this, in African cities economic and social disparities exacerbate health inequalities. The impact of air pollution of chemical and biological origin, which varies both spatially and temporally throughout urban centers, causes further health inequities and influences vulnerable populations. Findings from currently available works have revealed the following.
Exposure of human population to chemical and biological aerosols is of particular concern in Africa.Major chemical components of PM include carcinogenic PAHs and NPAHs and major the biological components in PM, including pathogenic fungi and bacteria, although information is scarce in Africa.The association of chemical and biological components of PM has been linked to synergistic health effects in other continents. However, the interrelationship of these factors is complex and deserves a comprehensive research in Africa.Chemical component of aerosols arises largely from automobiles and wood burning as the major sources of PAHs and NPAHs in Africa.Major knowledge gaps persist, particularly for the sub-Saharan region of Africa.

The total number of studies in Africa is extremely low and more are critically needed to better understand the contribution of both the biological and the chemical components of particulate matter to health outcomes in Africa. The limited funding and expertise in this field necessitates international and interdisciplinary collaboration.

## Figures and Tables

**Figure 1 ijerph-16-00941-f001:**
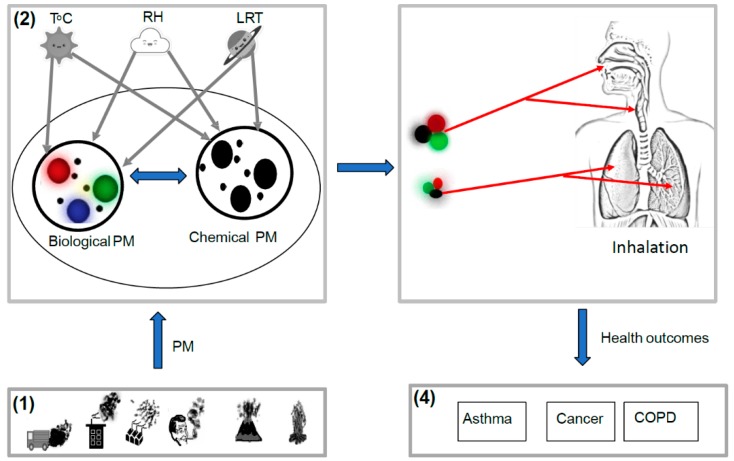
A schematic representation of the complex relationships between biological and chemical components of particulate matter (PM). (**1**) The sources of airborne PM; (**2**) the interaction of chemical and biological components of PM through the influence of T °C (temperature), RH (relative humidity), and LRT (long range transport); (**3**) routes of exposure to the mixture of PM_2.5_ (particulate matter with aerodynamic diameter less than 2.5 micrometers) that can enter the lungs and PM_10_ (particulate matter with aerodynamic diameter less than 10 micrometers) that are trapped in the nasopharyngeal from chemical and biological origins; and (**4**) possible health outcomes (Chronic Obstructive Pulmonary Diseases (COPD), asthma, and cancer).

**Figure 2 ijerph-16-00941-f002:**
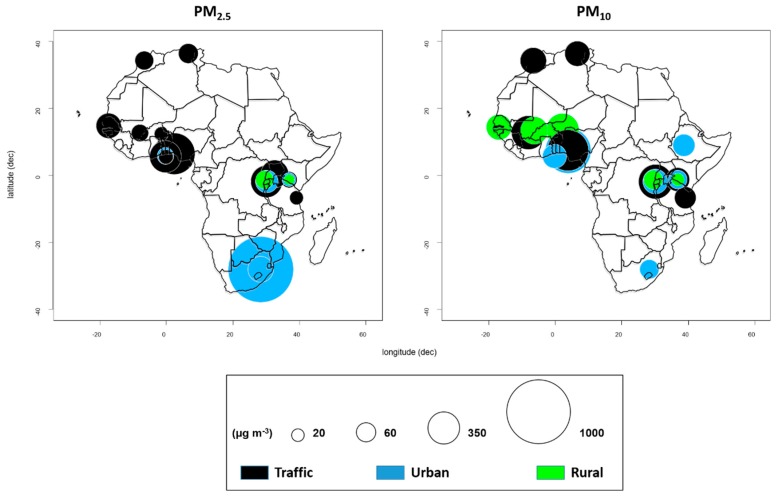
Ambient PM_2.5_ (particles less than 2.5 μm in diameter) (left), and ambient PM_10_ (particles less than 10 μm in diameter) (right), mean concentration as reported in studies from traffic (back color), urban background (blue), and rural site (green) in African countries such as Algeria [45], Benin [46], Burkina Faso [47], Ethiopia [48], Ghana [49,50], Kenya [51], Mali [52], Morocco [53], Niger [54], Nigeria [49], Rwanda [34], Senegal [54,55], South Africa [56,57], Tanzania [58], and Uganda [59].

**Figure 3 ijerph-16-00941-f003:**
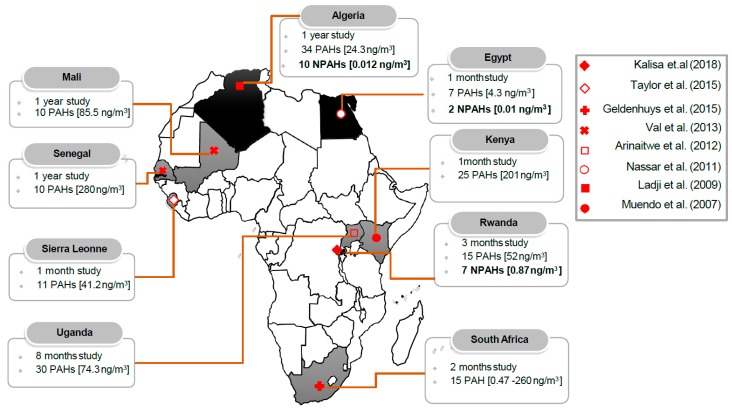
Map of Africa showing countries where studies on polycyclic aromatic hydrocarbons (PAHs) and their nitro-derivatives (NPAHs) in ambient air were conducted. The sampling duration, concentration of PAHs and NPAHs, and number of analyzed PAHs and NPAHs species; shaded black color indicates PAHs and NPAHs studies carried in Algeria [117], Egypt [116], and Rwanda [34] and shaded gray color indicates PAHs studies carried in Kenya [118], Mali [120], Sierra Leonne [121], Senegal [120], South Africa [119], and Uganda [122].

**Table 1 ijerph-16-00941-t001:** Summary of epidemiological and toxicological studies conducted in Africa on health effects of exposure to a mass concentration of ambient particulate matter size fraction. Particulate matter: ambient PM_2.5_ (particles less than 2.5 μm in diameter), ambient PM_10_ (particles less than 10 μm in diameter), TSP (total suspended particles).

Study	Study Location	Type of Study	Study Population	Statistical Analysis	PM Size Fraction	Association and Health Outcome
Mentz et al. [60]	Durban, South Africa	Longitudinal	N = 423 school children	Generalized estimating equation (GEE); 0–5 day lags; single lags and distributed lags	PM_10_ and PM_2.5_	Exposure to PM_10_ was associated with significantly increased occurrence of respiratory symptoms among children (cough, shortness of breath, and chest tightness).
Lin et al. [61]	South Africa, Ghana	Cross-sectional	N = 45,625, global aging and adult health	Logistic regression—3-level multilevel model	PM_2.5_	PM_2.5_ was found to be associated with overall disability and with cognition and mobility.
Makamure et al. [62]	Kwazulu-Natal, South Africa	Longitudinal/questionnaire	N = 71, Children ages 7–9	Linear multivariate	PM_10_ and PM_2.5_	Air pollution exposure results in increased expression of cluster of differentiation (CD14) in airway macrophages.
Ana et al. [63]	Ibadan, Nigeria	Cross-sectional	N = 140 ages 15–65 years	ANOVA and Spearman-rank correlation	PM_10_	Higher PM_10_ burden was observed to cause declining lung function.
Wichmann & Voyi [64]	Cape Town, South Africa	Case-crossover	N = 149,667 (RD = 13,439; CVD = 21,569; CVD = 7594)	Logistic regression	PM_10_	PM_10_ was associated with cardiovascular disease, respiratory disease, cerebrovascular disease, and mortality.
Mustapha et al. [65]	Ibadan, Nigeria	Cross-sectional	N = 1397 Schoolchildren (7–14 years)	Logistic regression	TSP, PM_2.5_ and PM_10_	Traffic pollution was associated with respiratory symptoms (wheeze, night cough, phlegm, rhinitis, and asthma in school children).
Kaphingst et al. [66]	Durban, South Africa	Longitudinal	N = 873 schoolchildren	Regression models	PM_10_ and PM_2.5_	Schoolchildren living near industries were more likely to develop asthma and airway hyperreactivity rather than those living far away from industries.

**Table 2 ijerph-16-00941-t002:** Summary of epidemiological and toxicological studies conducted in Africa on health effects of exposure to mass concentration of ambient particulate matter size fraction.

Reference	City, Country	Type of Site	PM Size	Main PAH Detected	Main NPAHs Detected	Source Identified	Association and Health Outcome
Kalisa et al. [34]	Kigali, Rwanda	Roadside/ambient air	PM_2.5_ and PM_10_	BPe, Phe, Flu, BaP, and BbF	9-NA, 2-NP+2-NFR, 6-NBaP	Wood burning and automobile emissions	The lifetime excess cancer risk exceeding the WHO guideline values and classified as definite risks.
Taylor et al. [121]	Western Sierra Leone	Residence/ambient air and indoor air	PM_2.5_	Phe, DBA, and BPe		Burning wood	PAHs bound PM_2.5_ from biomass fuel from kitchens continue to be hazardous for people of developing countries.
Geldenhuys et al. [119]	South Africa	Underground/ambient air	TSP	Pyr, Flu, and BaP		Diesel vehicle	Diesel exhaust emissions—recently confirmed as carcinogenic which is why the health of underground workers is of concern.
Val et al. [120]	Bamako, Mali	Desert area/ambient air	PM_10_	IDP, BPe, BbF, and BaP		Traffic, biomass burning, and dust	The population of Mali—highly exposed to toxic particulate pollution that could lead to strong adverse health effects.
Dieme et al. [55]	Dakar (Senegal)	Urban/ambient air	PM_2.5_	BbF, BPe, IDP, and BaP		Combustion of fossil fuels	PAH and Heavy metals in PM_2.5_ induced with dose-dependent toxicity, relying on inflammatory processes.
Hassan & Khoder [128]	Dokki, Egypt	Urban/ambient air	TSP	BbF, BPe, DBA, and Chr		Unburned fossil fuels and vehicle emissions	PAHs in the particulate phase in the ambient air posing a potential health risk for the population of Egypt.
Arinaitwe et al. [122]	Entebbe, Uganda	Watershed/ambient air	PM_2.5_	Phe, Flu, and Pyr		Combustion of petroleum and biomass burning	Population of Uganda is likely to be exposed to toxic PAHs bound PM_2.5_ from biomass burning.
Nassar et al. [116]	Great Cairo, Egypt	Traffic side/ambient air	TSP	Phe, Flu, BbF, and Chr	1-NP	Gasoline engine	PAHs and NPAHs with carcinogenic and/or mutagenic health effects detected in Greater Cairo.
Ladji et al. [117]	Algiers, Algeria	Suburban/ambient air	PM_10_	Acy, Phe, and BbF	9-NA, 2-NFR	Motor vehicles	The population of Algeria exposed to the occurrence of nicotine in particulates associated with PAHs.
Muendo et al. [118]	Nairobi, Kenya	Traffic/ambient air	PM_10_	Pyr, BbF, and BPe		Gasoline and diesel	Contribution of carcinogenic PAHs bound PM_10_ in Nairobi—approximately 30%.

Abbreviations of NPAH compounds: 9-nitroanthracene (9-NA), 2-nitropyrene (2-NP); 2-nitrofluoranthene (2-NFR), 1-nitroperylene (1-NP), 6-nitrochrysene, and 6-nitrobenz(a)pyrene (6-NBaP). Abbreviations of PAH compounds: Acenaphthylene (Acyl), phenanthrene (Phe), fluoranthene (Flu), pyrene (Pyr), benz(a)anthracene (BaA), chrysene (Chr), benzo(b)fluoranthene (BbF), benzo(a)pyrene (BaP), dibenz(a,h)anthracene (DBA), benz(g,h,i)perylene (BPe), and indeno (1,2,3-cd)pyrene (IDP). Particulate matter: PM _2.5_ (particles less than 2.5 μm in diameter), PM_10_ (particles less than 10 μm in diameter), TSP (total Suspended particle).

**Table 3 ijerph-16-00941-t003:** Summary of the available information on the types of study, biological pollutants analyzed (either singly or in combination with PM), study population and location, observed health effects, and the details of cited references.

Study	Study Location	PM Size	Biological Components Analyzed	Enumeration Techniques.	Dominant Species Identified	Association and Health Outcome
Abdel-Rahim et al. [138]	Assiut, Egypt	TSP	Fungi	Culture-dependent	*Chaetomium globosum*, *Aspergillus parasiticus*, *Penicillium oxalicum*, and *Setosphaeria rostrata*	The current study suggests that improvement of antimicrobial additives of paints may be a promising approach to reduce paint biodeterioration and, subsequently, air contamination of indoor environments.
Osman et al. [136]	Bolak, Egypt	>8 µm and <8 µm	Bacteria/Fungi	Culture-dependent	*Bacillus licheniformis*, Aspergillus, and Penicillium	Dust particles accumulated in air conditioning filters and floor surfaces and these would constitute important sources of airborne bacteria and fungi inside these hospitals.
Setlhare et al. [139]	South Africa	TSP	Bacteria/Fungi	Culture-dependent	Bacillus, Kocuria, Staphylococcus, Arthrobacter, Candida, Aureobasidium, Penicillium, and Phoma	Airborne bacteria and fungi that cause disease, especially in those populations with suppressed host immunity defenses in South Africa. Fungal genera identified (e.g., Candida), causes food spoilage and fungal infections in human
Rahoma [137]	Tobruk, Libya	0.2 µm	Bacteria/Fungi	Culture-dependent	*Bacillus thuringiensis* and *Cladosporium* sp. *Trichophyton* sp.	Inhalation of associated pathogenic viable microorganisms and chemical contaminants such as carcinogens and small particles may trigger other physiological reactions (e.g., asthma and cardiovascular events) in humans.
Kellogg et al. [140]	Bamako, Mali	TSP	Bacteria/Fungi	Culture-dependent	*Acinetobacter calcoaceticus*, *Bacillus mycoides*, *Bacillus pumilus*, *Bacillus subtilis*, and *Cladosporium cladosporioides*	Opportunistic human pathogens were isolated from air sample and could cause severe respiratory diseases

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
