# Peer review of "Chemical and Biological Components of Urban Aerosols in Africa: Current Status and Knowledge Gaps"

_ijerph, 2019, doi:10.3390/ijerph16060941_

Round 1
Reviewer 1 Report
REVIEW
Manuscript ID: ijerph-442544
General comments
The manuscript is well-constructed and has as its specific objective to highlight substantial gaps in the observation of pollution rates in Africa, specially PAHs/NPAHs and pathogenic microorganisms.
Given the relevance in terms of human health that the different species of pollutants have, the work fills a gap on the researches carried out on the African continent. As stated by the authors “the available data from biological composition associated with atmospheric PM comes mostly from Asia, Europe, and the United States”, while in Africa most countries have never programmed studies (very critical element).
The design is appropriate for IJERPH and the research question proposed should be of value and of sufficient interest to readers. The associated bibliography is very consistent and is overall suitable for this study type.
I propose few modifications or clarifications to the work, evaluating it in the complex deserving to be published.
###Abstract: line 25 “…… aerosols in disease causation….”. The term causation seems rather strong, it is more appropriate to use association.
### Table 2 - is not specified on which sample of the population the studies were carried out.
In addition, if possible, in Table 2 the authors could provide more information about statistical methods used in the cited studies. In addition, in table 2 the studies are classified by their typology (outdoor, indoor...). For studies carried out in indoor environments, it is unclear whether they refer to occupational indoor environments or other types of indoor environments.
### In Introduction add this reference (Review): Mastrangelo G., Fadda E. and Marzia V. Polycyclic Aromatic Hydrocarbons and Cancer in Man. Environmental Health Perspective, 1999, Vol. This Review report that “after 40 years of exposure, it involves a relative risk of 1.2-1.4 for lung cancer and 2.2 for bladder cancer”.
### Add and briefly comment on this reference: Rajendra KC, Shakti D. Shukla, Sanjay S. Gautam et al. The role of environmental exposure to non-cigarette smoke in lung disease. Clin Transl Med 2018; 7: 39. In particular, see Figure n. 1 of this reference.
Introduction - Chronic obstructive pulmonary disease (COPD). In Figure 1 and more broadly in the manuscript this disease does not appear even though it is very important.
### Figure 3: (Mali, Senegal and Sierra Leone…..2 days studies?? Are they meaningful monitoring studies? They really seem too short to include them in the review. Clarify.
### This reference (citation number 123) traces the state of the art on PAHs/NPAHs monitoring in Africa: “ Klánová, J.; Čupr, P.; Holoubek, I.; Borůvková, J.; Přibylová, P.; Kareš, R.; Ocelka, T. 764 Monitoring of persistent organic pollutants in Africa. Part 1: Passive air sampling across the continent in 2008. J. Environ. Monit. 2009, 11,1952.”
Are there no more recent studies? After 10 years no updates?
### African nations are currently 54. The number of States where monitors have been made are 13 (see Fig. 3). In the other 31 Nations there are no monitoring systems, although primitive or recent?
Author Response
Reviewer: 1
Manuscript ID: ijerph-442544
General comments
The manuscript is well constructed and has as its specific objective to highlight substantial gaps in the observation of pollution rates in Africa, specially PAHs/NPAHs and pathogenic microorganisms.
Given the relevance in terms of human health that the different species of pollutants have, the work fills a gap on the researches carried out on the African continent. As stated by the authors “the available data from biological composition associated with atmospheric PM comes mostly from Asia, Europe, and the United States”, while in Africa most countries have never programmed studies (very critical element).
The design is appropriate for IJERPH and the research question proposed should be of value and of sufficient interest to readers. The associated bibliography is very consistent and is overall suitable for this study type.
I propose few modifications or clarifications to the work, evaluating it in the complex deserving to be published
We thank the reviewer for the constructive comments and for appreciating the positive impact of the data presented in this manuscript to the scientific community. We have addressed the reviewer’s comments as follows:
Specific comments:
1. ###Abstract: line 25 “…aerosols in disease causation….”. The term causation seems rather strong, it is more appropriate to use association
We acknowledge the reviewer’s comment and in response to this, we have replaced causation with association (Line 25)
2. ### Table 2 - is not specified on which sample of the population the studies were carried out. In addition, if possible, in Table 2 the authors could provide more information about statistical methods used in the cited studies. In addition, in table 2 the studies are classified by their typology (outdoor, indoor...). For studies carried out in indoor environments, it is unclear whether they refer to occupational indoor environments or other types of indoor environments.
We have amended the table (now Table 1) in response to your suggestion, adding columns for Type of study, Population and Statistics.
3. ### In Introduction add this reference (Review): Mastrangelo G., Fadda E. and Marzia V. Polycyclic Aromatic Hydrocarbons and Cancer in Man. Environmental Health Perspective, 1999, Vol. This Review report that “after 40 years of exposure, it involves a relative risk of 1.2-1.4 for lung cancer and 2.2 for bladder cancer
We acknowledge the reviewer’s comment and the suggested reference was added in introduction section. (Line 71-72).
4. ### Add and briefly comment on this reference: Rajendra KC, Shakti D. Shukla, Sanjay S. Gautam et al. The role of environmental exposure to non-cigarette smoke in lung disease. Clin Transl Med 2018; 7: 39. In particular, see Figure n. 1 of this reference
Several sentences have been added on suggested reference (Line 44 - 47).
5. Introduction - Chronic obstructive pulmonary disease (COPD). In Figure 1 and more broadly in the manuscript this disease does not appear even though it is very important
We have amended Figure 1 and COPD was included as possible health outcome of exposure to association of chemical and biological aerosol. We have cited the work of Li et al 2018, Chen & Chen, 2018 and WHO, 2018 as an example of COPD as a resulting from exposure to PM .
6. ### Figure 3: (Mali, Senegal and Sierra Leone…..2 days studies?? Are they meaningful monitoring studies? They really seem too short to include them in the review. Clarify
We realised that there was an error in the sampling duration indicated in those three locations. We have corrected the information in Figure 3.
7. This reference (citation number 123) traces the state of the art on PAHs/NPAHs monitoring in Africa: “ Klánová, J.; Čupr, P.; Holoubek, I.; Borůvková, J.; Přibylová, P.; Kareš, R.; Ocelka, T. 764 Monitoring of persistent organic pollutants in Africa. Part 1: Passive air sampling across the continent in 2008. J. Environ. Monit. 2009, 11,1952. Are there no more recent studies? After 10 years no updates?
We agree with the reviewer that the reference mentioned is not the most recent. However, to the best our knowledge this is the only study found that has measured atmospheric PAHs in passive air samples in the African region. More recent studies available in the literature on PAHs did not meet the inclusion criteria since they have reported PAHs from soil and water instead of ambient air sample. The latest study that measured atmospheric PAHs in Africa was published by our research group last year (2018).
Kalisa, E.; Nagato, E. G.; Bizuru, E.; Lee, K. C.; Tang, N.; Pointing, S. B.; Hayakawa K.; Archer, S.D.J.; Lacap-Bugler, D. C. Characterization and Risk Assessment of Atmospheric PM2.5 and PM10 Particulate-Bound PAHs and NPAHs in Rwanda, Central-East Africa. Environ. Sci. Technol. 2018, 52, 12179–12187.
8. ### African nations are currently 54. The number of States where monitors have been made are 13 (see Fig. 3). In the other 31 Nations, there are no monitoring systems, although primitive or recent?
To the best of our knowledge there is no other study found that measure atmospheric PAHs/NPAHs in Africa or either primitive ways of measuring other than those provided. More studies available in the literature carried in Africa on PAHs did not meet the inclusion criteria since they have reported PAHs from soil and water instead of ambient air.
Reviewer 2 Report
Generally – please point out clearly which of the data presented in this paper is related to ambient (outdoor) air and which is related to indoor air. In Table 1 (2) page 5 results for both possible sorts of air pollution are presented.
All abbreviations must be explained in the text, compare for example “IDP” in Table 2 (3) page 12
I think, writing a review requires a particular resp. special attention to the cited literature. I am sorry, but I cann`t see this attention in the present paper.
Just to show some examples
1. United Nations. World Urbanization Prospects the Revision Highlights. Desa . 2007, 2, 883
2. World Health Organisation. Household air pollution and health . WHO Media centre. 2016 .
Both citations can`t be found as stated
3. Sigman, R.; Hilderink, H.; Delrue, N.; Braathen, N. A.; Leflaive, X. OECD Environmental Outlook to 2050. OECD. Environ. Outlook. 2012, 207–273.
Please cite this publication as: OECD (2012), OECD Environmental Outlook to 2050, OECD Publishing. http://dx.doi.org/10.1787/9789264122246-en (compare page 4 of the cited paper)
4) Why citing the secondary literature please cite the original WHO publication see
http://www.euro.who.int/__data/assets/pdf_file/0006/189051/Health-effects-of-particulate-matter-final-Eng.pdf?ua=1
5. International Agency for Research on Cancer. International Agency for Research on Cancer IARC Monographs on the Evaluation of Carcinogenic Risks to Humans. IARC. 2018, 1–118
References should be cited in the order like they show up in the text. This is not done throughout the present paper. Some examples:
Page 2 Lit [12 and 17] are not mentioned there.
Page 4 Lit [32,33,34,39, 40, 41,43,44,46] are not mentioned there
Fig 1 Please be careful with the wording “Death” - please describe more precisely- compare the original WHO-paper WHO Health effects of PM. Policy implications for countries in eastern Europe…
Line 59
The fact, that breathable air often contains PM is much elder, please compare for example - WHO Air quality guidelines for particulate matter, ozone, nitrogen dioxide and sulfur dioxide Global update 2005 Summary of risk assessment WHO/SDE/PHE/OEH/06.02
https://apps.who.int/iris/bitstream/handle/10665/69477/WHO_SDE_PHE_OEH_06.02_eng.pdf?sequence=1
Line 81 /82
a) “WHO regulatory limits” are “WHO guideline values”
b) Which information is limited- Indoor air resp. ambient air?
Line 91/92 Please explain the meaning of “Evidence sources [26,27]”
Line 95 f Please cite the source of this EPA statement
Line 101 Please point out where the statement that PM pollution is above the annual and the 24 hour mean is mentioned in Lit 28. As fare a I have seen Lit 28 deals with water pollution.
Line 102 ff Lit 30 is a WHO Guideline based on data from the 80Th and 90th and written in 2000 . I don`t believe a study from the year 2000 can be referred as “recently” in the year 2019. If citing this paper it should be cited correct. The evidence mentioned there refers to Europe and not to the world (globally). Please change this sentence.
In this context please note the first sentence of the abstract.
Line 107 Which study reports the fact, that there is one city that meets the PM10 guideline value? Lit ?
Line 119 f Four reviews on urban … have been publish [45,47{, where are the other two studies cited?
Line 121 f Lit 29 does not indicate an association of cooking and traffic PM emissions.
Line 123 Lit 35 says in its abstract “There is evidence for the contributions from biomass and traffic sources, and from geological and marine non-combustion sources to particle pollution”. So if citing this paper please cite all effects.
Line 127 – the table there is table 1
Table (page 5) – Lit Lin et al [47] the cited literature says more to the correlation of PM2.5 to health effects than mentioned / cited in this tabel
Line 116 and line 124 as well as Table 2 Please point out what are the reasons to choose the literature in Table 2 as a summarize of PM pollution in Africa. Why not citing literature 35 in this table?
Line 164
Lit 69 is just a secondary literature for this EPA statement – compare my remarks to line 95, please cite the original EPA statement.
Line 288
To which guide line value (ambient / indoor , guideline value 2006 or 2018 ) does this refers statement refer
Figure 3
Refer the information given in Fig. 3 to ambient or indoor air?
Fig 4
What`s TotalPAH? Sum of 6, 16 PAH? Which matrix? Ambient air indoor air?
Author Response
Reviewer 2
We acknowledge the constructive comments and suggestions of the reviewer, and we have made necessary amendments to the manuscript to address them.
Comments and Suggestions for Authors
Generally – please point out clearly which of the data presented in this paper is related to ambient (outdoor) air and which is related to indoor air.
We have made changes to the manuscript to illustrate whether the studies are from ambient air or indoor air, we have also summarised this information in supplementary table S1.
2. In Table 1 (2) page 5 results for both possible sorts of air pollution are presented. All abbreviations must be explained in the text, compare for example “IDP” in Table 2 (3) page 12.
We apologize for the omission. All PAH and NPAH abbreviations have been added in the manuscript where it is required and the full definition of any other abbreviation was provided at their first appearance.
3. I think, writing a review requires a particular resp. special attention to the cited literature. I am sorry, but I cann`t see this attention in the present paper.
Just to show some examples
1. United Nations. World Urbanization Prospects the Revision Highlights. Desa . 2007, 2, 883
2. World Health Organisation. Household air pollution and health . WHO Media centre. 2016
Both citations can`t be found as stated
4. Sigman, R.; Hilderink, H.; Delrue, N.; Braathen, N. A.; Leflaive, X. OECD Environmental Outlook to 2050. OECD. Environ. Outlook. 2012, 207–273
Please cite this publication as: OECD (2012), OECD Environmental Outlook to 2050, OECD Publishing. http://dx.doi.org/10.1787/9789264122246-en (compare page 4 of the cited paper)
We apologise for this shortcoming. We have double-checked all references to make sure they are well cited and can be found online.
5. Why citing the secondary literature please cite the original WHO publication see
http://www.euro.who.int/__data/assets/pdf_file/0006/189051/Health-effects-of-particulate-matter-final-Eng.pdf?ua=1
The primary literature has been added.
6. International Agency for Research on Cancer. International Agency for Research on Cancer IARC Monographs on the Evaluation of Carcinogenic Risks to Humans. IARC. 2018, 1–118.References should be cited in the order like they show up in the text. This is not done throughout the present paper. Some examples
Page 2 Lit [12 and 17] are not mentioned there.
Page 4 Lit [32,33,34,39, 40, 41,43,44,46] are not mentioned there
For comments 3 to 6.
We acknowledge the reviewer’s comment. We have cited reference consistently and all references were double-checked to make sure they appeared in the list of references. We also found an error in the citation software where it only showed the first and last number so some journals listed in the reference list did not show in the body of the text. This has been corrected in the present version.
7. Fig 1 Please be careful with the wording “Death” - please describe more precisely- compare the original WHO-paper WHO Health effects of PM. Policy implications for countries in eastern Europe
Figure 1 has been amended and the word ”Death” was replaced by chronic obstructive pulmonary disease (COPD).
8. Line 59
The fact, that breathable air often contains PM is much elder, please compare for example - WHO Air quality guidelines for particulate matter, ozone, nitrogen dioxide and sulfur dioxide Global update 2005 Summary of risk assessment WHO/SDE/PHE/OEH/06.02
We acknowledge the comment of the reviewer and we understand that breathable air does not only contain PM but also gases such as SO2, NOx, and VOC, which may react with other compounds present in the air to form fine particles. The current review only focuses on the associated PAHs/NPAHs and biological components of PM. We have excluded other gaseous air pollutants to put emphasis on PAHs and its nitrate derivatives. We have restructured some of the sentences to make this more explicit.
9. Line 81 /82
a) “WHO regulatory limits” are “WHO guideline values”
The sentence was modified to WHO annual and 24 hours guideline value of PM2.5 and PM10 (Line 94).
b) Which information is limited- Indoor air resp. ambient air?
Studies on characterization of PM components are very limited in Africa for both indoor and ambient air. The sentence has been modified. (Line 95-96)
10. Line 91/92 Please explain the meaning of “Evidence sources [26,27]”
The composition of PM vary substantially according to time, location, season, and climate, which results in spatial-temporal variation in characteristics, concentration and toxicity. In this sentence, we wanted to indicate that for controlling PM, it is very important to identify its actual source and the characterization of each PM components, which can then allow the evaluation of origin of each pollutant.
Line 95 Please cite the source of this EPA statement
We have added the reference for Environmental Protection Agency (EPA).
11. Line 101 Please point out where the statement that PM pollution is above the annual and the 24 hour mean is mentioned in Lit 28. As fare a I have seen Lit 28 deals with water pollution
We have amended the sentence to be more explicit and corrected the reference cited.
12. Line 102 ff Lit 30 is a WHO Guideline based on data from the 80Th and 90th and written in 2000. I don`t believe a study from the year 2000 can be referred as “recently” in the year 2019. If citing this paper it should be cited correct. The evidence mentioned there refers to Europe and not to the world (globally). Please change this sentence.
In this context please note the first sentence of the abstract.
We acknowledge the error in the reference cited. We have corrected the reference and have also omitted the word “worldwide” in the abstract. We have changed worldwide to low- and middle-income countries to be more specific.
13. Line 107 Which study reports the fact, that there is one city that meets the PM10 guideline value? Lit ?
Laako et.al 2018 reported the annual median concentration of PM10 (18.8 µg/m3) in Southern Africa, which was below WHO Guideline of PM10. Figure 2 and sentences were modified to be more clear.
Laakso, L.; Laakso, H.; Aalto, P. P.; Keronen, P.; Petäjä, T., Nieminen, T.; & Molefe, M. Basic characteristics of atmospheric particles, trace gases and meteorology in a relatively clean Southern African Savannah environment. Atmos. Chem. Phys. Discussions. 2008, 8(2), 6313-6353.
14. Line 119 f Four reviews on urban … have been publish [45,47{, where are the other two studies cited?
The citation for this sentence has been amended, (Line 157).
15. Line 121 f Lit 29 does not indicate an association of cooking and traffic PM emissions
The sentences were modified and the correct citations were added. (Line 158-164).
16. Line 123 Lit 35 says in its abstract, “There is evidence for the contributions from biomass and traffic sources, and from geological and marine non-combustion sources to particle pollution”. So if citing this paper please cite all effects
The sentence has been modified (Line159-163)
17. Line 127 – the table there is table 1
Table 2 has been changed to Table 1.
18. Table (page 5) – Lit Lin et al [47] the cited literature says more to the correlation of PM2.5 to health effects than mentioned / cited in this table.
The information in the table (now Table 1) has been modified to accurately represent the findings of the literature cited.
19. Line 116 and line 124 as well as Table 2 Please point out what are the reasons to choose the literature in Table 2 as a summarize of PM pollution in Africa. Why not citing literature 35 in this table?
We acknowledge the reviewer’s comment, and we have added a paragraph at the end of the introduction section to explain the inclusion criteria we used for this review.
Lit 35 did not meet the selection criteria. This particular literature was not included in the table.
20. Line 164
Lit 69 is just a secondary literature for this EPA statement – compare my remarks to line 95, please cite the original EPA statement.
This has been addressed.
21. Line 288
To which guide line value (ambient / indoor, guideline value 2006 or 2018) does this refers statement refer
We were referring to the WHO ambient air quality guidelines value (24-hour mean) of 2006. The sentences has been modified to clarify this point (Line 160).
22. Figure 3
Refer the information given in Fig. 3 to ambient or indoor air?
Data presented in Figure 3 are from ambient PM bound PAHs and NPAHs. The description of the figure has been modified.
23. Fig 4
What`s TotalPAH? Sum of 6, 16 PAH? Which matrix? Ambient air indoor air?
The total PAHs is referring to the summation of mean of each PAHs compound analysed in each country. The total PAHs in Figure 4 were identified in ambient air and the sentence has been modified to describe the sample.
Round 2
Reviewer 2 Report
First of all thank you for paying attention to my comments.
Nevertheless I have to make a few more comments
Line 131/132
“However, information on chemical and biological compositions of PM in both indoor and indoor air is still ..”
Sorry I don´t understand this part of this sentence. Please point out clearly what should be expressed
Line 524 Figure 4
1) Please change the way of citing in the figures subtext Klánová et al. (2009) is [123] as far as I have seen.
2) I´m sorry but I don`t have access to this literature (Klánová et al. (2009)) in the moment I have to make some basic comments to this figure. Even it is a correct citation, it is a matter of course that every sum depends on the number of numbers to be added. Thus it is obvious, that for the same indoor or ambient air situation there is a significant difference if 6, 7, 15 16 or 17 different PAH are summed up. This relevant information is not part of the presented figure 4. Thus either add this information or delete the figure.
3) The available abstract of this paper (Klánová et al. (2009 refers the passive sampling, in figure 4 several results for active sampling are presented, please refer the source of this results.
Line 642 Table 2 column – Type of site
Please add the information which matrix was analyzed - indoor air or ambient air,
for example for the first row - road side / ambient air
or the second row - residence / indoor air
and so on for the rest of the table
Author Response
First of all thank you for paying attention to my comments. Nevertheless, I have to make a few more comments.
We thank the reviewer for these positive comments and appreciate the acknowledgement of the noticeable improvement of the revised manuscript. We have addressed the additional comments as follows:
1. Line 131/132 “However, information on chemical and biological compositions of PM in both indoor and indoor air is still ..”Sorry I don´t understand this part of this sentence. Please point out clearly what should be expressed
The sentences have been modified. (Line 113-116)
2. Line 524 Figure 4
1) Please change the way of citing in the figures subtext Klánová et al. (2009) is [123] as far as I have seen.
2) I´m sorry but I don`t have access to this literature (Klánová et al. (2009)) in the moment I have to make some basic comments to this figure. Even it is a correct citation, it is a matter of course that every sum depends on the number of numbers to be added. Thus it is obvious, that for the same indoor or ambient air situation there is a significant difference if 6, 7, 15 16 or 17 different PAH are summed up. This relevant information is not part of the presented figure 4. Thus either add this information or delete the figure.
3) The available abstract of this paper (Klánová et al. (2009 refers the passive sampling, in figure 4 several results for active sampling are presented, please refer the source of this results.
We acknowledge the comments of the reviewer and in response, Figure 4 and cited references have been removed from the manuscript.
Line 642 Table 2 column – Type of site
Please add the information which matrix was analyzed - indoor air or ambient air, for example for the first row - roadside / ambient air or the second row - residence / indoor air and so on for the rest of the table.
The Type of site column in Table 2 has been modified.